# The Rapid Detection of *Salmonella enterica*, *Listeria monocytogenes*, and *Staphylococcus aureus* via Polymerase Chain Reaction Combined with Magnetic Beads and Capillary Electrophoresis

**DOI:** 10.3390/foods12213895

**Published:** 2023-10-24

**Authors:** Nodali Ndraha, Hung-Yun Lin, Shou-Kuan Tsai, Hsin-I Hsiao, Han-Jia Lin

**Affiliations:** 1Department of Bioscience and Biotechnology, National Taiwan Ocean University, Keelung 202301, Taiwan; nodali@email.ntou.edu.tw (N.N.); hungyun@mail.ntou.edu.tw (H.-Y.L.); 2Center of Excellence for the Oceans, National Taiwan Ocean University, Keelung 202301, Taiwan; 3BiOptic Inc., New Taipei City 23141, Taiwan; eric.tsai@bioptic.com.tw; 4Department of Food Science, National Taiwan Ocean University, Keelung 202301, Taiwan; hi.hsiao@email.ntou.edu.tw

**Keywords:** food safety, foodborne detection, PCR-based method, nucleic acid amplification, *Salmonella enterica*, *Listeria monocytogenes*, *Staphylococcus aureus*

## Abstract

Food safety concerns regarding foodborne pathogen contamination have gained global attention due to its significant implications. In this study, we developed a detection system utilizing a PCR array combined with an automated magnetic bead-based system and CE technology to enable the detection of three foodborne pathogens, namely *Salmonella enterica*, *Listeria monocytogenes*, and *Staphylococcus aureus*. The results showed that our developed method could detect these pathogens at concentrations as low as 7.3 × 10^1^, 6.7 × 10^2^, and 6.9 × 10^2^ cfu/mL, respectively, in the broth samples. In chicken samples, the limit of detection for these pathogens was 3.1 × 10^4^, 3.5 × 10^3^, and 3.9 × 10^2^ cfu/g, respectively. The detection of these pathogens was accomplished without the necessity for sample enrichment, and the entire protocols, from sample preparation to amplicon analysis, were completed in approximately 3.5 h. Regarding the impact of the extraction method on detection capability, our study observed that an automated DNA extraction system based on the magnetic bead method demonstrated a 10-fold improvement or, at the very least, yielded similar results compared to the column-based method. These findings demonstrated that our developed model is effective in detecting low levels of these pathogens in the samples analyzed in this study. The PCR-CE method developed in this study may help monitor food safety in the future. It may also be extended to identify other foodborne pathogens across a wide range of food samples.

## 1. Introduction

Bacterial food-borne diseases present a substantial global public health challenge, especially in developing nations. The World Health Organization (WHO) reports that food-borne or waterborne diarrhea causes approximately 2.2 million deaths worldwide each year [1]. The primary implicated microorganisms behind foodborne outbreaks are typically *Salmonella*, *Listeria*, and *Staphylococcus*. For instance, Dewey-Mattia et al. [2] reported that *Salmonella* spp., *L. monocytogenes*, and *S. aureus* caused 896, 191, and 35 foodborne outbreaks, respectively, in the United States from 2009 to 2015, which resulted in 23,662, 2378, and 1255 illnesses, respectively. The authors also reported that most of the infections were linked to the consumption of seafood and meat. In European countries, these pathogens caused 10,954, 90, and 1084 foodborne outbreaks, respectively, from 2016 to 2021, resulting in 72,954, 798, and 8948 illnesses, respectively [3]. In China, Chen et al. [4] reported that *Salmonella* and *S. aureus* caused 81 and 24 foodborne outbreaks, respectively, in Zhejiang Province from 2015 to 2020, which resulted in 1032 and 288 illnesses, respectively.

Poultry meat, including chicken meat, is known for being a rich source of essential nutrients, including protein, vitamins, minerals, niacin, amino acids, and fats. However, it can also serve as a potential carrier for foodborne pathogens like *Salmonella* spp., *L. monocytogenes*, and *S. aureus* [5,6,7,8,9,10,11]. In a recent study, Gonçalves-Tenório et al. [5] focused on analyzing the prevalence of pathogens in poultry meat across European countries using a meta-analysis approach. Their analysis, based on 78 published studies conducted in 21 European countries, revealed that the prevalence of *L. monocytogenes* and *Salmonella* spp. in poultry meat was 19.3% (95% CI: 14.4–25.3%) and 7.10% (95% CI: 4.6–10.8%), respectively [5]. Siriken et al. [6] collected 150 chicken samples from various supermarkets in Samsun, Turkey, and found that 42.7% of these samples contained *Salmonella* spp. Another study by Thung et al. [7], involving 120 chicken samples collected from wet markets and hypermarkets in Selangor, Malaysia, found that *Salmonella* Enteritidis and *Salmonella* Typhimurium were present in 6.7% and 2.5% of the samples, respectively, with the estimated quantities ranging from <3 to 15 MPN/g. Furthermore, Rortana et al. [8] collected 156 chicken meat samples from traditional markets in Cambodia, reporting that 40.4% and 46.2% of these samples contained *Salmonella* spp. and *S. aureus*, respectively. Additionally, in a set of 18 chicken meat samples collected from supermarkets, they did not detect any *S. aureus*, but 16.7% of them contained *Salmonella* spp. In terms of *L. monocytogenes*, recent studies have revealed its presence in chicken samples at rates of 45.0%, 46.7%, and 56.0% in Iran [9], Turkey [10], and Spain [11], respectively.

To enhance the control of microbial contamination in food products and reduce the occurrence of foodborne illnesses, rapid and accurate bacterial detection methods are, therefore, warranted. Although conventional culture-based methods have long been considered the “gold standard” for identifying pathogenic bacteria, they come with several drawbacks [12,13]. Typically, culture-based methods are time-consuming, labor-intensive, and less sensitive. The use of culture-based methods could also lead to potential false negative results, particularly when dealing with viable but non-culturable (VBNC) pathogens [12,13]. The utilization of polymerase chain reaction (PCR)-based methods has revolutionized pathogen detection by enabling faster, more sensitive, and more specific detection of pathogens in food samples [12,13].

However, the accuracy of PCR results relies, in part, on the quality and quantity of the sample. The analysis via PCR-based methods may produce false negative or false positive results [14,15]. False negative results may occur because of a failure of nucleic acid amplification caused by an insufficient quantity of bacterial cells and the presence of PCR inhibitors [14,15]. Relying on long enrichment times to produce a high number of bacterial cells in a particular sample can be disadvantageous since it fails to prevent food contamination from undesired bacteria. Several studies have reported that certain food and culture media components inhibit PCR [16,17]. False positive results may occur due to contamination during sample preparation, PCR amplification, or amplicon analysis [18]. Therefore, there is a high demand for a simple and selective food sample preparation method that involves the selective separation and concentration of the target bacteria from the food matrix, limits the amplification of DNA from viable cells, and minimizes the occurrence of contamination during the analysis.

Recently, there has been a rise in the production and commercialization of automated devices that use magnetic beads to minimize contamination during DNA extraction [19,20,21]. These devices can capture target bacteria selectively from different sample matrices. The magnetism of the beads can then be used to separate and concentrate the bacteria from large sample volumes using an external magnetic field, which could make the sample enrichment unnecessary as well as help eliminate PCR inhibitors [16,22]. After the amplification of nucleic acid, several researchers have used capillary electrophoresis (CE) [23,24] to improve the amplicon analysis as well as minimize the occurrence of post-PCR contamination. In this study, we aimed to develop a rapid detection method for detecting and identifying *S. enterica*, *L. monocytogenes*, and *S. aureus* in broth and chicken meats using a PCR array combined with an automated magnetic bead-based system and CE assay. While previous studies have explored the individual effectiveness of magnetic beads or CE in improving PCR performance for detecting foodborne pathogens in food samples [25,26,27,28,29,30], limited information is available regarding the overall effectiveness of integrating these technologies into the entire process of foodborne pathogen detection in poultry meat samples [31]. Chicken meat was used as the consumption of this food was expected to grow more quickly than any other significant meat [32,33].

## 2. Materials and Methods

### 2.1. Bacterial Strains, Media, and Culture Conditions

The bacterial species used in this study are purchased from the Bioresource Collection and Research Center (BCRC) Taiwan, including *S. enterica* BCRC 10747, *L. monocytogenes* BCRC 14845, and *S. aureus* BCRC 11863. *S. enterica* was cultured in tryptic soy broth (TSB) (HiMedia, Mumbai, India) and xylose lysine deoxycholate agar (Neogen, Lansing, MI, USA). *L. monocytogenes* was cultured in TSB with 0.6% yeast extract and nutrient agar (HiMedia). *S. aureus* was cultured in brain heart infusion (BHI) (HiMedia) broth and BHI agar (HiMedia). Each bacteria species was cultured in 10 mL broth or spread on an agar plate medium and incubated at 37 °C overnight. The stock for each bacteria species was preserved in aliquots of each culture with 25% glycerol and stored in a freezer at a temperature of −80 °C.

Each of the overnight bacteria cultures was centrifuged (3000× *g*, 15 min, 4 °C) and washed in phosphate-buffered saline (PBS). The bacteria pellets were then resuspended in PBS, and the optical density (OD) was adjusted to 1.0 to obtain 7.3 × 10^8^ of *S. enterica*, 6.7 × 10^8^ of *L. monocytogenes*, and 6.9 × 10^7^ cfu/mL of *S. aureus*. The number of bacterial cells was enumerated using the plate count method. Next, each bacteria strain was prepared in equal concentration (i.e., ~10^7^ cfu/mL). Thereafter, the mixture of bacteria strains was prepared by mixing the 3 target pathogens in equal volumes. The mixture of bacterial strains was serially diluted in volumes ranging from 10^0^ to 10^7^ CFU/mL separately.

### 2.2. Inoculation of Chicken Meat Samples

The chicken meat was purchased from a local slaughterhouse and subsequently divided into several parts, with each part weighing 10 g in a sterile plastic bag with a lateral filter (3M™, Saint Paul, MN, USA) and stored at −20 °C for overnight. To eliminate any background bacteria, the samples were sterilized using 6 kGy of gamma irradiation. The irradiation of samples was conducted at the laboratory of the National Atomic Research Institute located in Taoyuan, Taiwan. Under the guidelines developed by the Codex Alimentarius Commission, gamma ray irradiation of food at doses up to 10 kGy is considered safe for human consumption [34]. Regulations in the US and China permit the irradiation of frozen food up to 7 kGy and 8 kGy, respectively [35,36]. For our study, we used the dose of 6 kGy of gamma irradiation, as previous studies have shown that this level of irradiation can reduce pathogen levels in meat samples by more than six orders of magnitude [37,38,39]. Post-irradiation, all samples were stored in a freezer at −20 °C. To inoculate the samples, 100 µL of a mixture of bacterial suspensions with concentrations of 10^7^, 10^6^, 10^5^, 10^4^, 10^3^, and 10^2^ CFU/mL were applied to the surface of 10 g of chicken sample, resulting in bacterial concentrations of approximately 10^5^, 10^4^, 10^3^, 10^2^, 10^1^, and 10^0^ CFU/g. Control samples were also prepared without inoculation. Post-inoculation, the surface of each sample was gently massaged and air-dried in a laminar hood for 5 min. Thereafter, 90 mL of PBS was added to each sample and stomached at the highest speed for 1.5 min using a stomacher (BagMixer Interscience, Saint-Nom-la-Breteche, France) to obtain 1:10 homogenate. Subsequently, each homogenate was serially diluted in PBS, and the number of bacterial cells was determined using the plate count method. The suspension was plated on xylose lysine deoxycholate agar (HiMedia) for *S. enterica*, *Listeria* selective agar (HiMedia) for *L. monocytogenes* and mannitol salt agar (HiMedia) for *S. aureus*.

### 2.3. Genomic DNA Extraction

Figure 1 presents the general overview of PCR-CE developed in this study. To increase the concentration of bacterial cells, 1 mL of each bacterial suspension or stomached sample aliquot was centrifuged at 21,000× *g* for 1 min, and the supernatant was discarded. This process was repeated three times for each sample. Next, genomic DNA was extracted using an automated nucleic acid extractor with a nano-MB nucleic acid extraction kit (TANbead, Taipei, Taiwan). For comparison, we also extracted the DNA from stomached chicken samples using the spin column-based DNA extraction method. The DNA extraction via the spin column-based method was carried out using the Presto^TM^ Mini gDNA Bacteria Kit (Geneaid Biotech Ltd., New Taipei, Taiwan) according to the manufacturer’s instructions. The extracted DNA was dissolved in 100 µL Tris-EDTA buffer and then stored at −20 °C or subjected to PCR-CE assay immediately.

### 2.4. Optimization of the PCR-CE

Table 1 presents the sequences of the primers used in this study, which were obtained from published studies [40,41,42,43]. Simplex PCR was performed using Thermocycling (Biometra Tone, Analytik Jena, Jena, Germany). All oligonucleotides used in this research were synthesized using Genomics (New Taipei City, Taiwan). The annealing temperature and the PCR reagents were optimized to obtain high amplification efficiency for each gene in the PCR assay. PCR reagents were obtained from 6 different companies available in Taiwan, including 1 type of premix mixture and 5 types of HotStart Taq DNA polymerase (Table 2). DNA amplification was performed in a total volume of 25 µL of PCR mixture containing 2.5 µL of the DNA sample, 0.5 µL of forward primer, 0.5 µL of reverse primer, and various concentrations of master mix (Table 2). The concentration of each master mix was prepared based on the company protocols. Next, the amplified products were analyzed via capillary electrophoresis using Qsep100 Advance coupled with S2 Standard Cartridge (BiOptic Inc., New Taipei City, Taiwan). This CE system is equipped with a fluorescence detector and a separate voltage of 1~15 kV.

## 3. Results

### 3.1. Identification of Pathogens via the PCR-CE

Figure 2 presents the detection of *S. enterica*, *L. monocytogenes*, and *S. aureus* via the PCR-CE system in pure culture and chicken samples. The molecular sizes of the PCR products corresponding to the target pathogens were determined by analyzing the peak profiles of a 1000 bp DNA ladder. The concentrations of *S. enterica*, *L. monocytogenes*, and *S. aureus* were 7.3 × 10^5^, 6.7 × 10^5^, and 6.9 × 10^5^ cfu/mL, respectively, in PBS. In the chicken samples, the concentrations of these pathogens were 5.6 × 10^5^, 7.0 × 10^5^, and 6.3 × 10^5^ cfu/g, respectively. The optimum temperature cycle for the PCR assay was established as follows: initial denaturation at 95 °C for 10 min, denaturation at 95 °C for 30 s, annealing temperature at 57 °C for 30 s, extension at 72 °C for 50 s, and final extension at 72 °C for 5 min. The results showed that there was no amplification observed in the negative control (NTC) and non-target bacteria, indicating that the PCR-CE method developed in this study exhibited good specificity. The PCR products for *L. monocytogenes* (164 bp), *S. aureus* (210 bp), and *S. enterica* (392 bp) matched the expected sizes compared to the DNA-1000 ladder.

### 3.2. Sensitivity Evaluation of PCR-CE System in Pure Culture of Bacteria

Figure 3 shows the evaluation of the sensitivity of the PCR-CE system on the detection of pathogens in pure culture using the genomic DNA extracted via the auto-extractor system (magnetic bead-based method). A serial dilution was then prepared from 10^5^ to 10^0^ cfu/mL. The results showed that the limit of detection (LOD) was 7.3 × 10^1^ cfu/mL for *S. enterica*, 6.7 × 10^2^ cfu/mL for *L. monocytogenes*, and 6.9 × 10^2^ cfu/mL for *S. aureus*. We noted that there were some slight shifts in the signal due to the changes in the CE cartridge.

### 3.3. Evaluation of the Sensitivity of the PCR-CE System in Artificially Contaminated Samples

Figure 4 shows the evaluation of the sensitivity of the PCR-CE system on the detection of pathogens inoculated in chicken samples. The genomic DNA was extracted using magnetic bead-based and column-based methods for comparisons. The results showed that the peak value of each PCR product showed a downward trend as the concentration of bacteria decreased. This study found that the LOD for the magnetic bead-based method for *S. enterica*, *L. monocytogenes*, and *S. aureus* was 3.1 × 10^4^, 3.5 × 10^3^, and 3.9 × 10^2^ cfu/g, respectively (Figure 4A–C). No specific amplification peaks were observed in the NTC. For the column-based method, we found that these pathogens in chicken meats could be detected as low as 3.1 × 10^4^, 5.3 × 10^4^, and 6.1 × 10^4^ cfu/g, respectively.

### 3.4. Evaluation of PCR-CE Using Multiple Enzymes

Figure 5 shows the effect of PCR reagents on the detection of pathogens in pure culture and chicken samples. In our study, we selected six commercial kits available in Taiwan. *S. enterica* was used in the pure culture and inoculated chicken samples as this pathogen is frequently found in poultry products (the genomic DNA was extracted from 10^5^ cfu/mL each sample). Our results indicated that the analytical sensitivity was also influenced by the brand of the PCR reagents. PCR reagents from BiOptic, BioTools, ZymesetBiotek, and GMbiolab produced the expected peak for *S. enterica* (*invA* gene, 392 bp), while other brands (Invitrogen and BioHelix) did not. As shown in Figure 5, these PCR reagents have similar effects on the detection of pathogens in both the pure culture (Figure 5A) and chicken samples (Figure 5B).

## 4. Discussion

In recent years, the development and diversification of food supply chains have raised concerns about foodborne illnesses caused by pathogens, posing a significant threat to food safety. Fresh food from different markets, coupled with complex food processing techniques, increases the likelihood of multiple pathogen contamination. Common foodborne pathogens, such as *S. enterica*, *S. aureus*, and *L. monocytogenes*, are responsible for a range of foodborne diseases. In Taiwan, Lai et al. [44] reported that these pathogens caused annual cases of 185,977, 432, and 17 foodborne illnesses based on surveillance data from 2012 to 2015. In another study, Yu et al. [45] revealed that *Salmonella* and *S. aureus* accounted for 14% and 19% of foodborne diseases in Taiwan based on surveillance data from 2014 to 2018. Hence, the development of rapid and accurate detection and identification of these pathogens in food is, therefore, crucial to effectively control food poisoning and prevent its further dissemination.

Several techniques have been developed to identify pathogenic bacteria in food and water. While culture-based methods have been commonly employed, they suffer from limitations such as time-consuming processes, potential false negative results, and labor-intensive procedures [12]. Serotyping is another widely used method for detecting and distinguishing pathogens; however, this approach has its drawbacks, including the inability to serotype a small percentage (between 5% and 8%) of isolates and potential errors in typing due to the loss of surface antigens [46,47]. Recent advancements in technology have prompted researchers to explore molecular biology methods and novel nanomaterials for pathogen detection [12,48]. Examples of these molecular methods include PCR-based methods, loop-mediated isothermal amplification (LAMP), recombinase polymerase amplification (RPA), rolling circle amplification (RCA), and DNA microarray analysis, among others [12,48]. In our study, the PCR technique was employed to amplify nucleic acid due to its reputation as one of the most reliable and accurate methods for pathogen detection. We acknowledge that various PCR-based detection methods have seen advancements over the last decade, with qPCR remaining the most commonly used approach [14]. However, as outlined in our prior review [14], each method has its own set of advantages and disadvantages. In our present study, we propose an alternative solution for PCR-based detection. Our current PCR-CE system not only offers relative cost-effectiveness compared to other PCR-based techniques but also offers the potential for portability [49]. This feature makes it a valuable tool for on-site pathogen detection. Additionally, our research demonstrated that the integration of an automated magnetic bead-based DNA extraction system with a PCR array and CE assay led to a rapid pathogen detection process, reduced contamination risks, and high-resolution analysis results.

Before the amplification process, genomic DNA from the target pathogens is typically extracted using a column-based DNA extraction method [50]. However, this method has its limitations. Firstly, it can be both labor-intensive and time-consuming due to its multi-step nature, involving binding, washing, and elution. These numerous steps contribute to extended handling time, rendering it less suitable for high-throughput scenarios [50]. Furthermore, the column-based approach may lead to lower DNA yield and purity, particularly when dealing with samples containing minimal DNA content or complex matrices [50,51]. In response to these challenges, researchers have devised an innovative alternative for isolating genomic DNA from samples—the magnetic bead-based method [51]. This technique offers several distinct advantages. Notably, it significantly enhances the speed and simplicity of the extraction process. By reducing the number of procedural steps, magnetic bead-based extraction expedites DNA isolation. Moreover, its compatibility with automation makes it ideal for processing a considerable number of samples simultaneously, making it particularly attractive for high-throughput laboratory settings. Additionally, the magnetic bead-based method often yields DNA samples of higher purity and improved recovery rates. In our study, the DNA from the bacterial cells was extracted using magnetic beads in an automated DNA extractor system. This extractor system only required 20–30 min of sample pre-treatment and approximately 1 h of DNA extraction. Compared to the column-based extraction method, the application of this automated machine based on magnetic bead-based separation exhibits a similar result and sometimes an improvement in detection capability by at least 10 times (Figure 4). Prior research has indicated that coating magnetic beads with antibodies could enhance the efficiency of DNA extraction [52,53,54]. However, the costs associated with acquiring or developing the necessary antibodies impede the widespread adoption of this technique [14]. Nevertheless, in our study, the peak signals generated from DNA extracted using the magnetic bead-based method (Figure 4A–C) were generally stronger than those produced using the column-based method (Figure 4D–F). This signifies that the automated DNA extractor system based on magnetic bead separation facilitates more efficient DNA extraction while concurrently reducing the presence of inhibitors, thereby optimizing the nucleic acid amplification process in the PCR.

Table 3 presents the detection limits of the PCR array combined with CE technology, comparing the results from previous studies with those from this present study. Zhou et al. [55] used multiplex PCR combined with a CE system to detect *Salmonella*, *E. coli* O157:H7, *L. monocytogenes*, *S. aureus*, *Shigella* spp., and *C. jejuni* in broths and found that their system could detect these pathogens at concentrations of 4.2 × 10^2^, 9.3 × 10^1^, 3.1 × 10^2^, 2.7 × 10^2^, 8.5 × 10^1^, and 6.6 × 10^1^ cfu/mL, respectively, without the pre-enrichment of the samples. In their protocol, the entire pathogen detection process could be completed within 4 h, including 1 h for DNA extraction, 2 h for PCR amplification, 45 min/row for CE separation, and 10 min for interpretation [55]. In another study, Ruan et al. [56] adopted a combination of duplex PCR and the CE system to identify *Cronobacter* spp. in broth, and their approach successfully detected this pathogen at a concentration of 1.6 × 10^1^ cfu/mL without requiring sample pre-enrichment. Regarding the detection of pathogens in food samples, Alarcón et al. [27] demonstrated that the CE system, in conjunction with a multiplex PCR array, could detect *S. aureus*, *L. monocytogenes*, and *Salmonella* spp. in raw beef at concentrations as low as 2.6 × 10^3^, 5.7 × 10^2^, and 7.9 × 10^2^ cfu/mL, respectively, without the need for enrichment. In their system, amplicon analysis via the CE system completed the analysis within 25 min. In terms of food safety management, allowing the detection of pathogens without requiring sample enrichment would provide early information to food managers so that necessary preventive action can be taken in a timely manner. In terms of protecting public health, such early information would help food safety authorities in the decision-making process to prevent the occurrence of food poisoning.

In our system, we used a simplex PCR to amplify the nucleic acid from the extracted DNA of *S. aureus*, *L. monocytogenes*, and *Salmonella* cultured in broth and inoculated in chicken meat. The inoculation of these pathogens in chicken meat was aimed to mimic the real situation. The detection of these pathogens using the CE system was completed within 4 min/row (Table 4). The PCR-CE system developed in our lab could detect these pathogens as low as 7.3 × 10^1^ for *S. enterica*, 6.7 × 10^2^ for *L. monocytogenes*, and 6.9 × 10^2^ for *S. aureus*, without the need for enrichment (Figure 3). In chicken meat samples, these pathogens could be detected as low as 7.3 × 10^4^, 6.7 × 10^3^, and 6.9 × 10^2^ cfu/mL, respectively, without the need for enrichment (Figure 4A–C). The less sensitive detection of these pathogens in chicken meat samples may be attributed to the presence of inhibitors in the chicken meat suspension [57]. These inhibitors could have potentially reduced the effectiveness of magnetic beads in capturing bacterial cells during DNA extraction or inhibited the amplification of target pathogens during the PCR process [58]. Magnetic beads may not efficiently capture bacteria in food samples that contain complex matrices with various components, including fats, proteins, and other particles, as they can get entangled or adsorb non-target substances, reducing the effectiveness of bacterial isolation [57,58]. A previous study has mentioned that the presence of fat and protein can inhibit nucleic acid amplification during the PCR process [57]. Additionally, other compounds used in DNA extraction, such as detergents, lysozyme, NaOH, and alcohols, can also have inhibitory effects [57]. We noticed that several studies have attempted to separate bacterial cells from food matrices by applying the principle of buoyant density gradient centrifugation [59,60,61]. This method involves the utilization of substances such as Percoll to effectively isolate bacterial cells from food matrices. However, the practical implementation of this technique is hindered by the requirement to prepare solutions with varying density gradients and the need for specialized tools and high-speed centrifugation equipment. Furthermore, the application of this method would be time-consuming and raise the risk of cross-contamination during the transfer of samples from one tube to another. Therefore, future studies must focus on reducing or eliminating the impact of these inhibitors to enhance the performance of nucleic acid amplification.

Furthermore, the adoption of the PCR-CE system in our study eliminates the requirement for an electrophoresis gel, a common component in traditional PCR methods. CE proves highly effective in segregating DNA fragments based on their size and charge [24,62,63]. In CE, a narrow capillary tube is filled with a conductive buffer solution. The application of an electric field across the capillary prompts the migration of the negatively charged DNA fragments [24,62,63]. Smaller fragments migrate swiftly, covering more distance within the capillary, whereas larger ones lag behind [24,62,63]. A detector records their migration time and peak intensity, facilitating the precise analysis and quantification of PCR products. Additionally, this system minimizes cross-contamination during amplicon analysis, a crucial consideration due to its potential to yield false positive results [13,14,15]. Our approach eliminates the need to transfer the amplicon from the PCR tube, allowing direct application in the CE system. This streamlined process significantly reduces the risk of cross-contamination and enhances analysis reliability. Despite the advantages of CE in DNA fragment separation, there is a need for the full integration of PCR arrays and CE technology, along with the refinement of software tailored to the PCR-CE system, to enhance pathogen detection efficiency.

We observed that as the bacterial concentration decreased, there was a corresponding increase in the occurrence of non-specific signals. These signals did not appear in samples with higher bacterial concentrations (i.e., 10^5^ cfu per mL or g, Figure 2). Actually, the occurrence of non-specific signals is a common issue encountered when working with PCR products [64,65,66,67]. This phenomenon may arise due to the amplification of unintended DNA fragments. Previous studies have suggested the use of organic molecules such as dimethyl sulfoxide (DMSO), glycerol, polyethylene glycol, formamide, and betaine to prevent the amplification of unintended DNA fragments [64,66,67,68]. However, in our study, the application of these organic molecules did not improve PCR amplification. Nevertheless, the occurrence of non-specific signals became less significant when a capillary electrophoresis system was employed, as presented in this study (Figure 3 and Figure 4). Capillary electrophoresis offered high-resolution detection, making it easier to identify non-specific signals, thereby reducing their impact on the analysis and ensuring the accuracy and reliability of the obtained results.

In terms of the formulation of PCR reagents from different brands, our study confirmed that it can affect the amplification of nucleic acid and thus affect the amplicon analysis. A previous study conducted by Lin et al. [49] observed that the formulation of PCR reagents obtained from various manufacturers had an impact on amplicon analysis. It is worth noting that Lin et al.’s study focused on detecting viruses in shrimp samples [49]. They discovered that PCR reagents manufactured by ThermoFisher, Biotools, and BiOptic enabled the detection of the target viruses in their samples, whereas the reagent from Takara (Takara Bio Inc., Tokyo, Japan) did not yield successful results. In this study, the effectiveness of the PCR-CE system was also dependent on the choice of PCR reagent. Our observation showed that two out of six types of reagents tested failed to enable the amplification of the nucleic acid during the PCR process. These findings suggest that caution should be exercised when using the PCR-CE system developed in this study, as different formulations of PCR reagents can impact the PCR process. Manufacturers may formulate their reagents according to their technologies, which may not work well when combined with technologies developed by other manufacturers.

Finally, the PCR-CE system that we have developed in this study demonstrated the capability to detect and identify the target pathogens within approximately 3.5 h, thereby improving upon the 4–5 h required in conventional PCR methods and the even lengthier 94 h necessitated in culture-based techniques (Table 4). This acceleration in the processing time can be attributed to the elimination of the requirement for sample enrichment and the use of CE, which facilitates the rapid separation of DNA fragments. The rapid detection of foodborne pathogens is crucial for effective food safety management [13,14]. It enables timely responses, minimizes the impact of outbreaks, and ultimately protects public health by ensuring that consumers can confidently enjoy the food they consume.

One limitation of this study is the inability of PCR-CE to differentiate between signals originating from viable cells and DNA released from dead cells. A positive signal could originate from non-viable cells, potentially resulting in false positive results. Such outcomes could have financial and legal implications for food business operators. One potential solution to address this concern is the use of RNA as an indicator [69]. However, RNA is generally less stable than DNA and can degrade quickly when exposed to environmental factors, such as temperature, pH, and enzymatic activity [70]. Another approach involves the application of viability dyes, such as propidium monoazide (PMA), DyeTox13, or thiazole orange monoazide (TOMO) [14,71]. Previous studies have indicated that these dyes could inhibit signals arising from dead cells [72,73]. However, the effectiveness of these dyes may be influenced by the food matrices and bacterial species [71]. Thus, future studies are suggested to explore the effectiveness of these dyes in combination with the PCR-CE assay for detecting live foodborne pathogens in various food samples.

## 5. Conclusions

In conclusion, the utilization of an automatic DNA extractor in conjunction with a PCR-CE assay, developed in this study, presents a faster and more cost-effective approach to detecting *S. enterica*, *L. monocytogenes*, and *S. aureus* in the pure culture and chicken meat. The entire process, encompassing automatic DNA extraction, PCR, and CE, can be accomplished in under 4 h. By directly integrating the automated magnetic bead-based DNA extraction system with the PCR array and CE technology, the risk of contamination is minimized. These combined technologies hold the potential to enhance food safety monitoring and offer a practical and indispensable application to ensure food safety in the future. Additionally, the developed method not only proves effective in detecting low levels of these pathogens in the samples analyzed in this study but can also be extended to identify other foodborne pathogens across a wide range of food samples.

## Figures and Tables

**Figure 1 foods-12-03895-f001:**
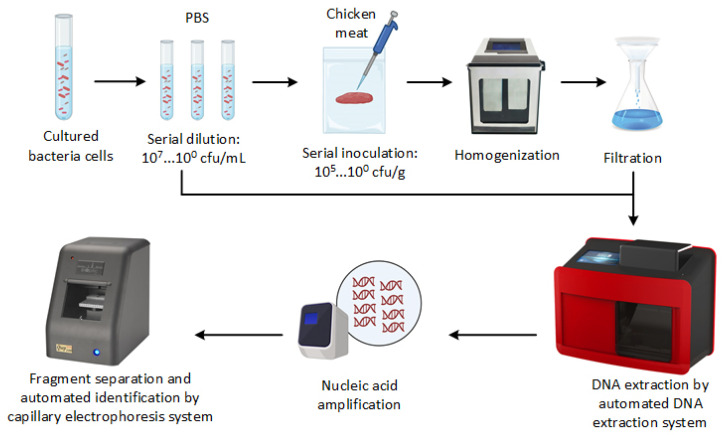
Schematic overview of the development of PCR-CE system for detecting pathogens.

**Figure 2 foods-12-03895-f002:**
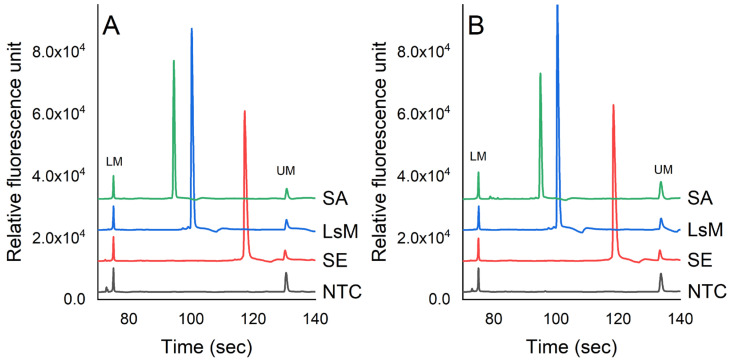
Detection of pathogens in (**A**) PBS and (**B**) chicken samples using PCR-CE system. The concentrations of *S. enterica* (*SE*), *L. monocytogenes* (*LsM*), and *S. aureus* (*SA*) were 7.3 × 10^5^, 6.7 × 10^5^, and 6.9 × 10^5^ cfu/mL, respectively, in PBS. In chicken samples, the concentrations of these pathogens were 5.6 × 10^5^, 7.0 × 10^5^, and 6.3 × 10^5^ cfu/g, respectively (LM = lower bound marker, UM = upper bound marker).

**Figure 3 foods-12-03895-f003:**
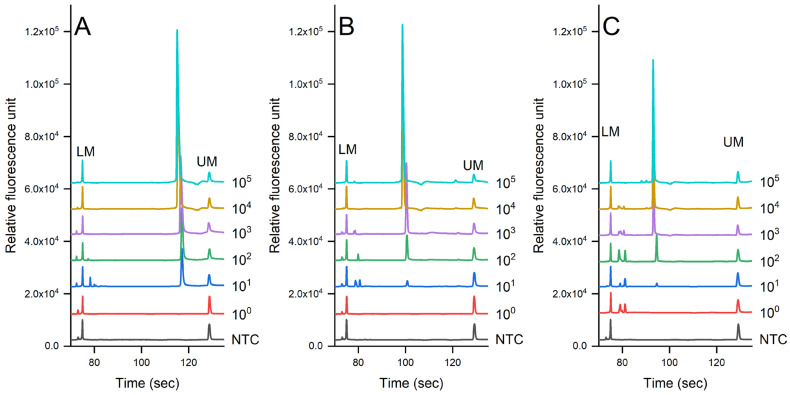
Sensitivity evaluation of (**A**) *S. enterica*, (**B**) *L. monocytogenes*, and (**C**) *S. aureus* in pure culture detected via PCR-CE systems (Serial dilution of bacterial concentration was prepared from 10^0^ to 10^5^ cfu/mL, NTC = non-template control).

**Figure 4 foods-12-03895-f004:**
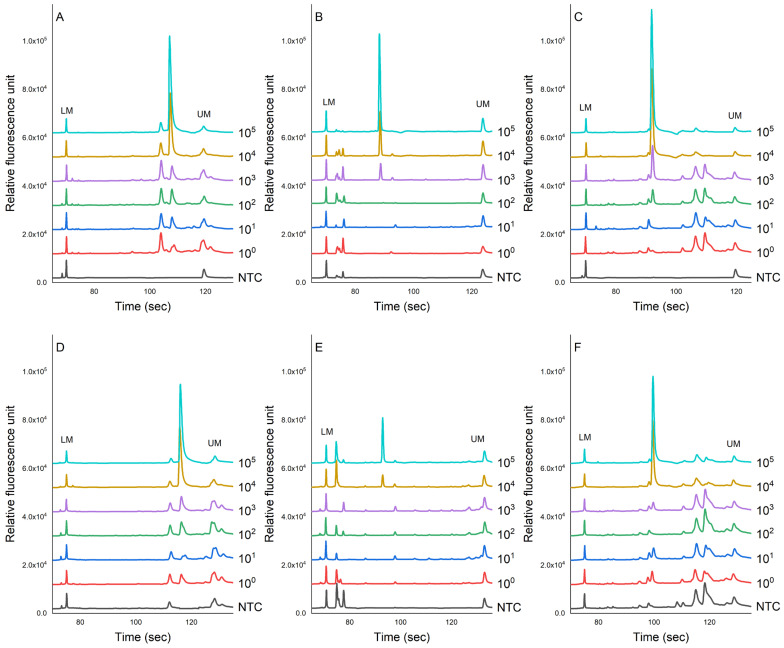
The sensitivity evaluation of the PCR-CE system for detecting pathogens, namely (**A**) *L. monocytogenes*, (**B**) *S. enterica*, and (**C**) *S. aureus*, inoculated in chicken meat samples. Two different extraction methods were employed: the magnetic bead-based method (**A**–**C**) and the column-based method (**D**–**F**). NTC = non-template control).

**Figure 5 foods-12-03895-f005:**
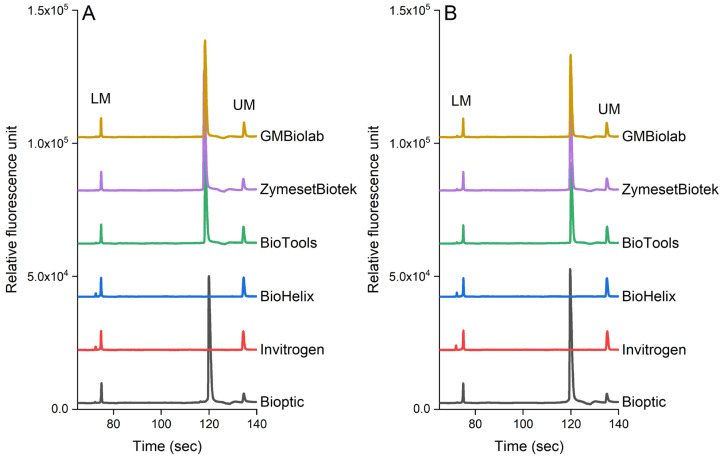
The performance of the PCR-CE system using various commercial enzymes in (**A**) pure culture and (**B**) chicken samples. The genomic DNA was extracted using a magnetic bead-based method from 7.3 × 10^5^ cfu/mL and 5.6 × 10^5^ of *S. enterica* in pure culture and chicken samples, respectively.

**Table 1 foods-12-03895-t001:** Oligonucleotide primers and probes used in the polymerase chain reaction assays.

Target Pathogens	Nucleotide Sequence	Target Gene	Size (bp)	Ref
*S. aureus*	F: AAA TTA CAT AAA GAA CCT GCG AC	*nuc*	164	[40]
	R: GCA CTT GCT TCA GGA CCA TA			
*L. monocytogenes*	F: CGC AAC AAA CTG AAG CAA AG	*hly*	210	[40]
	R: TTG GCG GCA CAT TTG TCA C			
*S. enterica*	F: GCC GCC AAA CCT AA	*invA*	392	[42]
	R: CTG CTA CCT TGC TGA TG			

**Table 2 foods-12-03895-t002:** PCR mixture using various polymerase enzymes.

Enzyme	Trade Name	Source	Volume (µL)						
			10× Taq Buffer	10 mM dNTP	Hot Start Taq	F-Primer	R-Primer	DNA Template	Water
GMbiolab	GenHot Taq DNA Polymerase	Gmbiolab Co., Ltd., Taipei, Taiwan	2.5	0.5	0.50	0.5	0.5	2.0	18.50
ZymesetBiotek	Hot start Taq DNA polymerase	ZymesetBiotek, Taipei, Taiwan	2.5	0.5	0.25	0.5	0.5	2.0	18.75
BioTools	TOOLS Hotstart Polymerase	Biotools, New Taipei City, Taiwan	2.5	0.5	0.50	0.5	0.5	2.0	18.50
BioHelix	nanoTaq Hot-Start DNA Polymerase	Bio-Helix Co., Ltd., New Taipei City, Taiwan	2.5	0.5	1.25	0.5	0.5	2.0	17.75
ThermoFisher	Invitrogen™ Platinum™ Taq DNA Polymerase ^1^	ThermoFisher, Waltham, MA, USA	2.5	0.5	0.10	0.5	0.5	2.0	18.15
BiOptic	DirectGO^TM^ PreMix-CE ^2^	BiOptic Inc., New Taipei City, Taiwan	Premix: 12.5			0.5	0.5	2.0	9.50

^1^ Note: 0.75 µL of Mg^2+^ was added to the mixture as this chemical was not included in the buffer. ^2^ PCR reagent from Bioptic was provided in a ready-to-use master mix containing reaction buffer, dNTPs, glycerol, PCR enhancers, and Hot Start Taq DNA polymerase as a 2-fold concentration.

**Table 3 foods-12-03895-t003:** Comparison of employing PCR coupled with CE array in detecting foodborne pathogens in broth and food samples.

Species Detected	Sample	Detection Limit, Direct	Detection Method	Reference
*L. monocytogenes*, *Salmonella* and *S. aureus*	Stomached raw beef filtrate	Lm: 5.7 × 10^2^, Sal: 7.9 × 10^2^, Sa: 2.6 × 10^3^ cfu/mL	Multiplex PCR	[27]
*Salmonella*, *E. coli* O157:H7, *L. monocytogenes*, *S. aureus*, *Shigella* spp. and *C. jejuni*	Broth	Sal: 4.2 × 10^2^, Ec: 9.3 × 10^1^, Lm: 3.1 × 10^2^, Sa: 2.7 × 10^2^, Sh: 8.5 × 10^1^, and Cj: 6.6 × 10^1^ cfu/mL	Multiplex PCR	[55]
*Cronobacter* spp.	Broth	Cr: 1.6 × 10^1^ cfu/mL	Duplex PCR	[56]
*S. enterica*, *L. monocytogenes*, and *S. aureus*	Broth	Sal: 7.3 × 10^1^, Lm: 6.7 × 10^2^, Sa: 6.9 × 10^2^ cfu/mL	Simplex PCR	This study
*S. enterica*, *L. monocytogenes*, and *S. aureus*	Stomached raw chicken filtrate	Sal: 7.3 × 10^4^, Lm: 6.7 × 10^3^, Sa: 6.9 × 10^2^ cfu/g	Simplex PCR	This study

**Table 4 foods-12-03895-t004:** Comparison of the time spent for pathogen identification between the culture-based method and PCR-CE method developed in this study.

Steps	Culture-Based Method ^1^	General PCR	PCR-CE Method
Sample preparation and dilution	~10 min	~30 min	~30 min
Pre-enrichment	~18 ± 2 h	−	−
Selective enrichment	~24 ± 3 h	−	−
Plating out	~24 ± 3 h	−	−
Confirmation ^2^	~24 ± 2 h	−	−
Serotyping	4–6 h	−	−
DNA isolation by commercial kit	−	~1 h (manual)	~1 h (automatic)
Nucleic acid amplification	−	~2 h	~2 h
Agarose preparation	−	~10 min	−
Amplicon analysis	−	~30 min	~4 min
Total spent time	~94 h (3.9 days)	~4 h and 10 min	~3 h and 34 min

^1^ The culture-based method for *Salmonella* adopted from the International Organization for Standardization protocol ISO 6579-1:2017. ^2^ Confirmation by API 20E (11iome’rieux) biochemical assays.

## Data Availability

Data are available on request from the authors.

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
