# Peer review of "The Rapid Detection of Salmonella enterica, Listeria monocytogenes, and Staphylococcus aureus via Polymerase Chain Reaction Combined with Magnetic Beads and Capillary Electrophoresis"

_foods, 2023, doi:10.3390/foods12213895_

Round 1

Reviewer 1 Report

Dear editors and authors

1-The introduction in the manuscript needs to add paragraphs about the presence of types of pathogenic bacteria in foods, and this is the subject of the study, so I suggest you read and review some previous research, such as

-Niamah, A. K. (2012). Detected of aero gene in Aeromonas hydrophila isolates from shrimp and peeled shrimp samples in local markets. The Journal of Microbiology, Biotechnology and Food Sciences, 2(2), 634.‏

-Riu, J., & Giussani, B. (2020). Electrochemical biosensors for the detection of pathogenic bacteria in food. TrAC Trends in Analytical Chemistry, 126, 115863.‏

2-Authors must write a clear purpose for the manuscript. At the end of the introduction, there is a purpose, the purpose of the current study must be written.

3-Why is there no chapter on statistical analysis in work methods? Statistical analysis is very important in such studies.

4-The conclusions contain many results, and this is not permissible. These results must be removed and the conclusions rewritten again.

Reviewer 2 Report

Review report

I sincerely appreciate the opportunity to review the manuscript entitled “Rapid detection of foodborne pathogens by PCR combined with magnetic beads and capillary electrophoresis. This article describes usage of PCR - capillary electrophoresis in detection of some (three) pathogens in broth and contaminated chicken. Despite its scientific quality and soundness, the authors need to resolve several issues.

Major Issues

- Major issue pertains to the fact that positive PCR signal (amplicon) does not necessarily indicate a "live" bacterial cell. Thus, a positive signal might be obtained from the non-viable cells causing false-positive results and incurring costs or legal activities against the food-business operator. Solution would be to use RNA as an indicator or any chemical treatment to exclude signals from dead bacterial cells. That is a reason why an overnight incubation is required in many PCR applications. Authors never mentioned this possibility and given its significance it is necessary to address this.

- Some figures are presented in the results as approximate (literally). I think this is not very precise, and it is preferable to mention the exact numbers.

- The discussion, although fairly written, requires a thorough revision, as it is mainly a repetition of concepts already known or presented in results, without much explanation what significance this approach may have compared to the e.g. multiplex qPCR (or similar more widespread techniques), since financial effects for sure are not a matter of concern.

- I am worried about novelty level this manuscript brings. Mentioned technique has already been described 20 years ago (Alarcon et al., 2004; Leader et al., 2009). Next, a similar research was further described by Zhang et al. (2018) "Simultaneous detection of three foodborne pathogenic bacteria in food samples by microchip capillary electrophoresis in combination with polymerase chain reaction" or Chen et al (2021) "Rapid Analysis for Staphylococcus aureus via Microchip Capillary Electrophoresis" and He et al. (2020) "Detection of four foodborne pathogens based on magnetic separation multiplex PCR and capillary electrophoresis".

Comments to be addressed

Line 103-104. Describe in detail process of gamma radiation and why did you choose 6 kGy?

Line 108. Why authors did not test contaminated samples in order to have exact figures in contamination level? In this way, contamination level is just assessed mathematically instead of empirically.

Line 120. Which column-based method did you choose? Please, describe in more details.

Line 127. Correct "Analytikjena" to "Analytik Jena".

Line 131. Correct "HotStart Tag" to "HotStart Taq".

Line 132. Did you mean "25 uL"? Please, make a correction.

Line 133. This is ambiguous "1 μL of each primer", since you used 0.5 + 0.5 uL of each primer (fwd+rev). Rephrase the sentence.

Line 152. This is not inline with Material and Methods: "and 10e7 cfu/g for chicken samples" because previously (in Line 107) authors stated that the peak concentration was 10e5 cfu/g in meat. Please explain?

Line 182-186. Presented results do not show any benefit or increased sensitivity of the PCR-CE method vs. column-based method, since LOD's are exactly the same for both methods!

Line 235-236. I do not agree with these claims since multiplex qPCR has become cheap and there is no any post-PCR setup. qPCR machines are more versatile compared to capillary fluorescence detectors.

Round 2

Reviewer 2 Report

Authors corrected majority of issues.